# PGE$_2$ accounts for bidirectional changes in alveolar macrophage self-renewal with aging and smoking

Loka R Penke[1],*, Jennifer M Speth[1],*, Christina Draijer[1], Zbigniew Zaslona[1], Judy Chen[2,4], Peter Mancuso[2,3], Christine M Freeman[1,2,6], Jeffrey L Curtis[1,2,7], Daniel R Goldstein[2,4,5], Marc Peters-Golden[1,2]

Alveolar macrophages (AMs) are resident immune cells of the lung that are critical for host defense. AMs are capable of proliferative renewal, yet their numbers are known to decrease with aging and increase with cigarette smoking. The mechanism by which AM proliferation is physiologically restrained, and whether dysregulation of this brake contributes to altered AM numbers in pathologic circumstances, however, remains unknown. Mice of advanced age exhibited diminished basal AM numbers and contained elevated PGE$_2$ levels in their bronchoalveolar lavage fluid (BALF) as compared with young mice. Exogenous PGE$_2$ inhibited AM proliferation in an E prostanoid receptor 2 (EP2)-cyclic AMP-dependent manner. Furthermore, EP2 knockout (EP2 KO) mice exhibited elevated basal AM numbers, and their AMs resisted the ability of PGE$_2$ and aged BALF to inhibit proliferation. In contrast, increased numbers of AMs in mice exposed to cigarette smoking were associated with reduced PGE$_2$ levels in BALF and were further exaggerated in EP2 KO mice. Collectively, our findings demonstrate that PGE$_2$ functions as a tunable brake on AM numbers under physiologic and pathophysiological conditions.

## Introduction

Alveolar macrophages (AMs) are the primary resident innate immune cells of the pulmonary alveoli. AMs recognize and clear inhaled pathogens and particulates, catabolize surfactant, and orchestrate the initiation and resolution of inflammation, which are critical to maintain lung homeostasis. AMs differ from other tissue macrophages in numerous respects; these differences have been attributed, at least in part, to the unique microenvironment of the alveolar space in which they reside (Hussell & Bell, 2014).

Tissue macrophages were long considered to arise predominantly from circulating monocytes. However, over the last decade, it has become well-accepted that at steady state, many resident tissue macrophages, including AMs, arise from embryonic precursors and maintain their numbers via self-renewal (Tarling et al, 1987; Guilliams et al, 2013; Hashimoto et al, 2013). Proliferative renewal is especially important in AMs because of their remarkable longevity in the lung (Maus et al, 2006). However, the regulation of proliferation in resident tissue macrophages, including AMs, remains poorly understood. It is worth noting that Gata6, a transcription factor implicated in the renewal of peritoneal macrophages (Rosas et al, 2014), is not expressed in AMs (Draijer et al, 2019). Nevertheless, AMs clearly proliferate in response to colony-stimulating factors GM-CSF and M-CSF (Chen et al, 1988; Draijer et al, 2019), with mTORC1 (Deng et al, 2017) and the transcription factors Bhlhe40 and Bhlhe41 (Rauschmeier et al, 2019) being responsible for this process. MafB, another transcription factor that is necessary for maintenance of AM numbers (Sato-Nishiwaki et al, 2013), is also expressed at very low levels in AMs (Hamada et al, 2020).

Common host-intrinsic and environmental factors have been linked to alterations in AM numbers in vivo. For example, reduced AM numbers contribute to the impaired pulmonary host defense associated with advanced age (Higashimoto et al, 1993; Wong et al, 2017). Conversely, an increased number of AMs associated with cigarette smoking (CS) (Hornby & Kellington, 1990; Barbers et al, 1991; Suzuki et al, 2020) have been implicated in the development of chronic obstructive pulmonary disease. Although these observations clearly establish that AM self-renewal is a tunable phenomenon, a unifying mechanism for such bidirectional alterations in AM numbers is lacking. Moreover, independent of recognized mechanisms promoting proliferation, the potential role of endogenous brakes on this process remains uncertain. Here, we used murine models of both aging and CS to reveal the importance of endogenous PGE$_2$ synthesis and signaling via the E prostanoid 2

[1]Division of Pulmonary and Critical Care Medicine, Department of Internal Medicine, University of Michigan Medical School, Ann Arbor, MI, USA   [2]Graduate Program in Immunology, University of Michigan Medical School, Ann Arbor, MI, USA   [3]Department of Nutritional Sciences, University of Michigan School of Public Health, Ann Arbor, MI, USA   [4]Division of Cardiology, Department of Internal Medicine, University of Michigan Medical School, Ann Arbor, MI, USA   [5]Institute of Gerontology, University of Michigan, Ann Arbor, MI, USA   [6]Research Service, Veterans Affairs Ann Arbor Healthcare System, Ann Arbor, MI, USA   [7]Medical Service, Veterans Affairs Ann Arbor Healthcare System, Ann Arbor, MI, USA

Correspondence: petersm@umich.edu
*Loka R Penke and Jennifer M Speth contributed equally to this work

receptor (EP2) as a novel brake on AM proliferative self-renewal that is itself subject to bidirectional modulation during perturbations.

# Results

## AM proliferative capacity is reduced in aged mice in association with increased levels of $PGE_2$ within the lung

Aging reduces the functions (Aprahamian et al, 2008; Boe et al, 2017; Li et al, 2017) and numbers (Zissel et al, 1999; Wong et al, 2017) of AMs in both mice and humans. We analyzed AM numbers in the bronchoalveolar lavage fluid (BALF) of 18–22-mo-old mice compared with 6–8-wk young controls and found a significant reduction in AMs with aging (Fig 1A). These data confirm previous findings in which AMs were enumerated from mouse lung digests rather than BALF (Wong et al, 2017).

Reduced AM numbers with aging could result from cell-intrinsic or extrinsic factors. To evaluate the role of intrinsic factors, we isolated AMs from both young and aged mice and measured their proliferation after stimulation with the important AM mitogen GM-CSF (Hashimoto et al, 2013). Compared with young AMs, the proliferative capacity of aged AMs in response to GM-CSF was significantly reduced (Fig 1B). In parallel, aged AMs exhibited a similar, significant reduction in the expression of the cell cycle gene *Cyclin B1* compared with young AMs (Fig 1C). As extrinsic factors may shape functional changes in AMs (McQuattie-Pimentel et al, 2019 *Preprint*), we next evaluated the contribution of microenvironmental alterations in the aged lung to AM proliferation. To test this, we measured AM proliferation in young AMs cultured with GM-CSF in the presence of cell-free BALF from either young or aged mice. Whereas AM proliferation was intact when cells were cultured with BALF from young mice, it was diminished when they were cultured with BALF from aged mice (Fig 1D). These data suggest the presence of both cell-intrinsic defects and suppressive factors in the alveolar microenvironment that contribute to impaired AM proliferation associated with aging.

We recently reported that AMs from aged mice exhibited increased eicosanoid gene expression (Wong et al, 2017). We therefore used lipidomics to obtain an unbiased evaluation of lipid mediator content in BALF from both young and aged mice. BALF from aged mice contained altered levels of a variety of lipid mediators (Fig 1E); these include increases in those derived from the arachidonate cyclooxygenase ($PGE_2$ and 12-hydroxyheptadecatrienoic acid [12-HHT]) and lipoxygenase (leukotriene $E_4$ and 12-hydroxyeicosatetraenoic acid [12-HETE]) metabolic pathways and decreases in the linoleic acid oxidation derivative 13-hydroxyoctadecadienoic acid [13-HODE]. Of these lipid mediators, the greatest increment in BALF from aged relative to young mice was observed for $PGE_2$. We used ELISA to confirm a significant elevation in $PGE_2$ within the BALF of an independent and larger cohort of aged mice (Fig 1F). To determine whether AM numbers and $PGE_2$ levels within the lungs of individual mice were related, correlation analysis was performed in a separate cohort of both young and aged mice. Indeed, there was a moderate but significant inverse correlation between AM numbers and $PGE_2$ levels of individual mice (Fig 1G).

## CS exposure induces AM proliferation concurrent with decreased $PGE_2$ production

In both human smokers and animal models of CS exposure, increased AM numbers (Hornby & Kellington, 1990; Barbers et al, 1991; Suzuki et al, 2020) accompany and have been implicated in a variety of forms of lung pathology and dysfunction (Jayes et al, 2016; Hu et al, 2019; Suzuki et al, 2020). However, there is conflicting evidence as to whether this reflects proliferation of resident AMs or recruitment of circulating monocytes (Golde et al, 1974; Hornby & Kellington, 1990; Hodge et al, 2007; Perez-Rial et al, 2013). We therefore enumerated resident AMs and assessed their proliferation in vivo in response to CS exposure (2 h/d for 7 d). Consistent with some previous studies, the total number of lavaged cells was significantly increased in mice exposed to 7 d of CS as compared with control mice exposed to room air (Fig 2A). This increase in total lavaged cells was accompanied by a significant increase in the frequency of resident AMs as identified by two independent sets of markers ($CD11b^-CD11c^+$ and $MerTK^+CD64^+$) (Fig 2B). There were no significant differences in the number of $CD11b^+CD11c^-Ly6G^+$ neutrophils or $CD11b^-CD11c^+CD103^+$ dendritic cells (Fig 2C), suggesting that these populations did not contribute to increased cell numbers in CS-exposed mice. There was a nonsignificant trend in numbers of $CD11b^+CD11c^-Ly6C^+$ recruited monocytes in CS-exposed mice (Fig 2C); this is unlikely to be meaningful, however, given their very low absolute numbers relative to numbers of resident AMs. To confirm that this observed increase in AMs was due to enhanced proliferation, we measured proliferation via 5-ethynyl-2′-deoxyuridine (EdU) incorporation within resident AMs during CS exposure. $CD11c^+$ AMs isolated from the BALF of CS-exposed mice displayed a significant increase in EdU incorporation compared with cells from air-exposed mice (Fig 2D). Taken together, the results in this model of CS exposure indicate that the increase in AM numbers is due to increased AM proliferation, rather than an influx of circulating inflammatory monocytes.

Alveolar epithelial cells comprise the alveolar surface, and like AMs (Balter et al, 1989), they have the capacity to synthesize $PGE_2$ (Chauncey et al, 1988). Of note, ex vivo $PGE_2$ production by AMs was reduced in human smokers compared with nonsmokers (Balter et al, 1989) and in vitro CS exposure reduced $PGE_2$ synthesis by cultured airway epithelial cells obtained by brushing of human subjects (Zhang et al, 2011). We therefore hypothesized that the increase in AM numbers and proliferation in smoked mice may be associated with decreased $PGE_2$ levels on the alveolar surface. Indeed, CS exposure resulted in a time-dependent decrease in $PGE_2$ levels within the lung over 7 d, and this decrease persisted with continuous CS exposure for as long as 8 wk (Fig 2E).

## $PGE_2$ inhibits mitogen-induced AM proliferation via EP2-cAMP signaling

The above data establish that $PGE_2$ bioavailability within the lung is inversely correlated with the diametrically opposed abnormalities of AM numbers and proliferation in two models of abnormal lung immune function: aging and smoking. We next directly assessed the in vitro effect of $PGE_2$ on AM proliferation and interrogated its operative signaling mechanisms. Treatment of naïve AMs with 1 $\mu M$

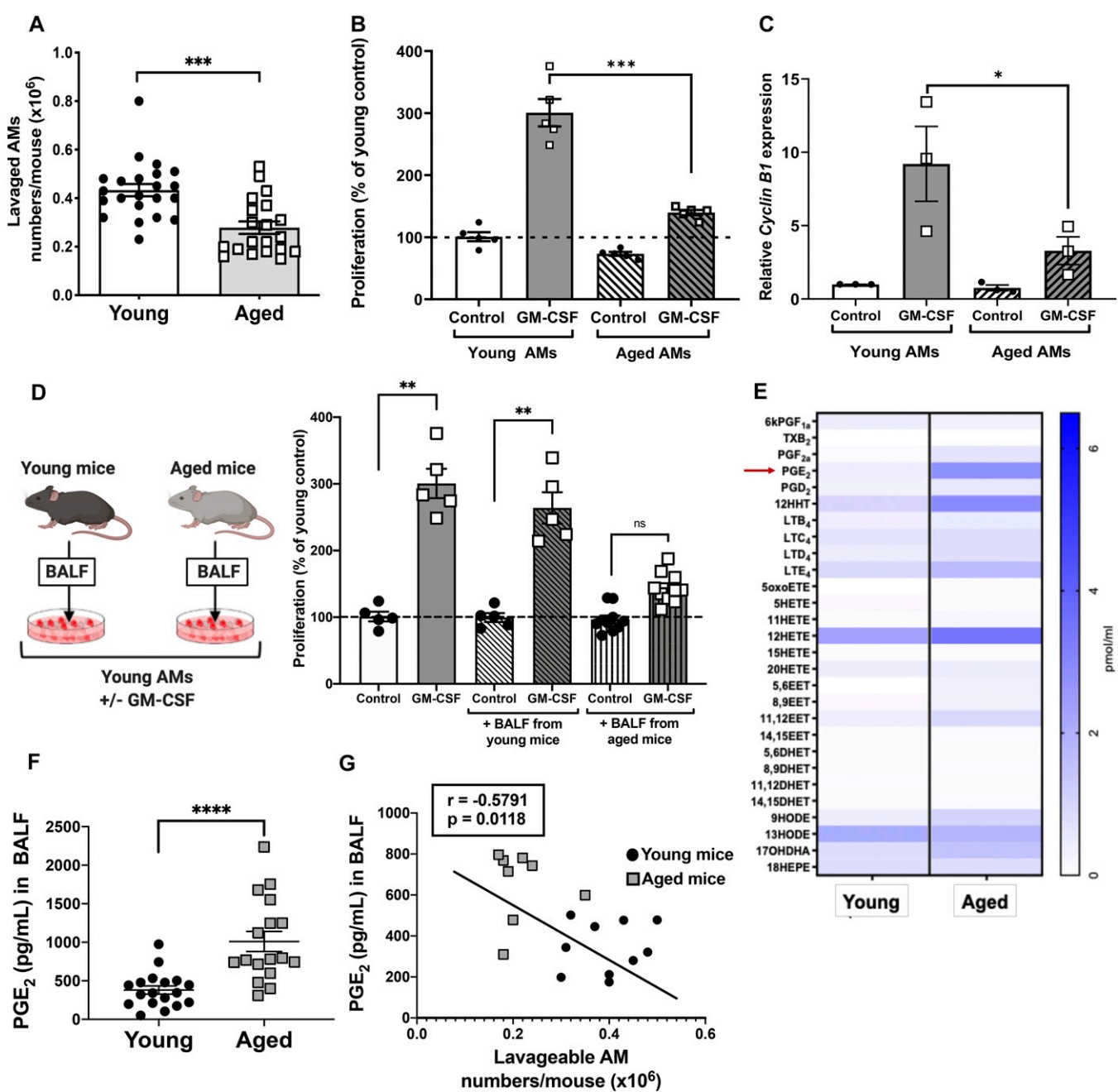

**Figure 1. Alveolar macrophage (AM) proliferation is decreased in aged mice in association with elevated levels of PGE₂.**
**(A)** Absolute numbers of AMs enumerated from BALF of young (6–8 wk old) and aged (18–22 wk old) mice (n = 20–22 mice). **(B)** Primary AMs isolated from young and aged mice were treated with mouse GM-CSF (10 ng/ml) and incubated for 5 d and subjected to CyQuant proliferation assay measuring total cellular DNA (n = 5 mice). **(C)** Expression of *Cyclin B1* mRNA measured by RT-PCR in young and aged primary AMs treated with GM-CSF for 48 h (n = 3 separate experiments). **(D-left panel)** Schematic depicting young and aged BALF swap experiment. **(D-right panel)** Proliferation of young AMs incubated for 5 d in the presence of young or aged BALF, or medium alone, with or without GM-CSF (n = 5–9 mice). **(E)** Heat map of lipids that were elevated in BALF from both young (n = 4 mice) and aged (n = 3 mice) mice. **(F)** Levels of PGE₂ from young or aged BALF quantified by ELISA (n = 17–18 mice). **(G)** Pearson's correlation analysis between AM numbers and PGE₂ levels in matched BALF samples from individual mice (n = 18 mice; 10 young, 8 aged). **(A, B, C, D, E)** Results in (B, C, D) are values expressed relative to those in unstimulated young AMs, and all data represent the mean ± SEM of the values indicated by individual symbols; ns, non-significant, *$P < 0.05$, **$P < 0.01$, ***$P < 0.001$; $t$ test (A, E) or one-way ANOVA followed by Sidak's test for multiple comparisons (B, C, D).

PGE₂ markedly reduced the ability of both GM-CSF and M-CSF to elicit AM proliferation (Fig 3A) and to induce expression of proliferation-associated genes (Fig 3B). We previously reported that among the four G protein–coupled receptors for PGE₂, EP2 is responsible for mediating most of its inhibitory effects on AM functions (Aronoff et al, 2004; Medeiros et al, 2009). Treatment with the selective EP2 agonist butaprost mimicked the ability of PGE₂ to inhibit AM proliferation in response to both GM-CSF and M-CSF (Fig

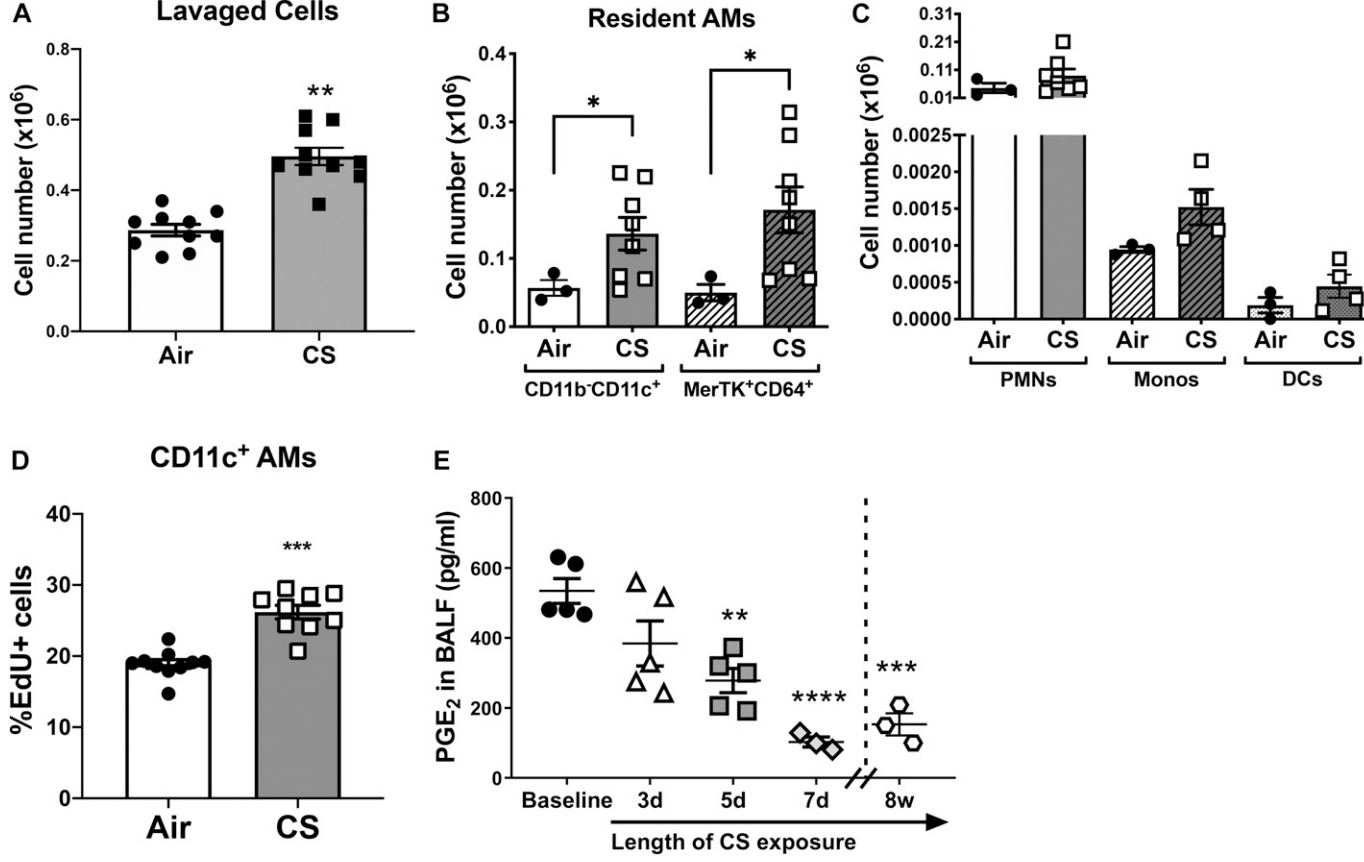

**Figure 2. Alveolar macrophage (AM) proliferation is reduced in cigarette smoking (CS)-exposed mice concurrent with decreased PGE$_2$ levels in the lung.**
**(A)** Absolute numbers of lavaged cells in BALF collected from mice exposed to air or CS for 7 d (n = 10 mice). **(B)** Absolute numbers of CD11b$^-$CD11c$^+$ and MerTK$^+$CD64$^+$ resident AMs isolated from lung digests of mice exposed to air or CS (n = 3–8 mice), as measured by flow cytometry. **(C)** Absolute numbers of CD11b$^+$CD11c$^-$Ly6G$^+$ neutrophils, CD11b$^+$CD11c$^-$Ly6C$^+$ monocytes, and CD11b$^-$CD11c$^+$CD103$^+$ dendritic cells isolated from lung digests of mice exposed to air or CS (n = 3–8 mice), as measured by flow cytometry. **(D)** Frequency of EdU$^+$CD11c$^+$ resident AMs isolated from the BALF of air or CS-exposed mice measured by flow cytometry. Mice were injected with EdU (1 mg/kg) 1 d before CS exposure and every other day for 7 total d (n = 9–10 mice). **(E)** Levels of PGE$_2$ from BALF of air or CS-exposed mice at baseline, 3 d, 5 d, 7 d and 8 wk (n = 3–5 mice). **(A, B, C, D, E)** Results are shown as mean ± SEM; *$P < 0.05$, **$P < 0.01$, ***$P < 0.001$; ****$P < 0.0001$; $t$ test (A, D) or one-way ANOVA followed by Sidak's test for multiple comparisons (B, C, E).

3C). EP2 is a stimulatory G protein–coupled receptor that signals by activating adenylyl cyclase to increase intracellular production of cAMP (Aronoff et al, 2004). Forskolin, a direct activator of adenylyl cyclase, exerted similar inhibitory effects as PGE$_2$ and butaprost on mitogen-induced proliferation (Fig 3C).

The effects of cAMP signaling on AM functions are mediated by two distinct downstream cAMP effector molecules, PKA and guanine nucleotide exchange protein directly activated by cAMP-1 (Epac-1) (Cheng et al, 2008). Treatment with cAMP analogs that are selective agonists for either PKA (6-bnz-cAMP) or Epac-1 (8-pCPT-2′-O-Me-cAMP) both resulted in significant inhibition of AM proliferation in response to GM-CSF (Fig 3D). Collectively, these data suggest that the inhibitory effects of PGE$_2$ on AM proliferation proceed via an EP2-cAMP signaling pathway. Both cAMP effectors are capable of contributing to this suppressive effect.

#### Endogenous EP2 signaling is necessary for PGE$_2$ restraint of AM numbers and proliferation in vitro and in vivo

To validate that endogenous EP2 signaling restrains AM proliferative self-renewal, we used mice with a global deletion of the EP2

receptor (EP2 KO). The number of AMs lavaged from naïve EP2 KO mice was significantly higher than that from wild-type mice at baseline (Fig 4A). In addition, AMs from EP2 KO mice displayed an exaggerated proliferative capacity in response to GM-CSF stimulation (Fig 4B).

It was possible that the increased AM renewal in EP2 KO mice could reflect, at least in part, a stimulatory effect of PGE$_2$ via the now-unchecked actions of signaling via alternative PGE$_2$ receptors, particularly the known stimulatory receptors EP1 or 3. We therefore tested the direct effects of exogenous PGE$_2$ on proliferation of EP2 KO AMs in the presence or absence of GM-CSF. As expected, the ability of PGE$_2$ to inhibit AM proliferation in response to GM-CSF was lost in EP2 KO AMs (Fig 4C) confirming that PGE$_2$ acts via EP2 to attenuate this process. Moreover, PGE$_2$ had no effect—stimulatory or otherwise—on either basal or GM-CSF–stimulated proliferation of EP2 KO AMs (Fig 4C). These data indicate that the increased AM renewal in EP2 KO mice is explained entirely by removal of the PGE$_2$-EP2 brake, rather than by any stimulatory actions of PGE$_2$ mediated via other EP receptors.

We next sought to determine the role of PGE$_2$/EP2 signaling on abnormal AM proliferation and cell numbers demonstrated in the

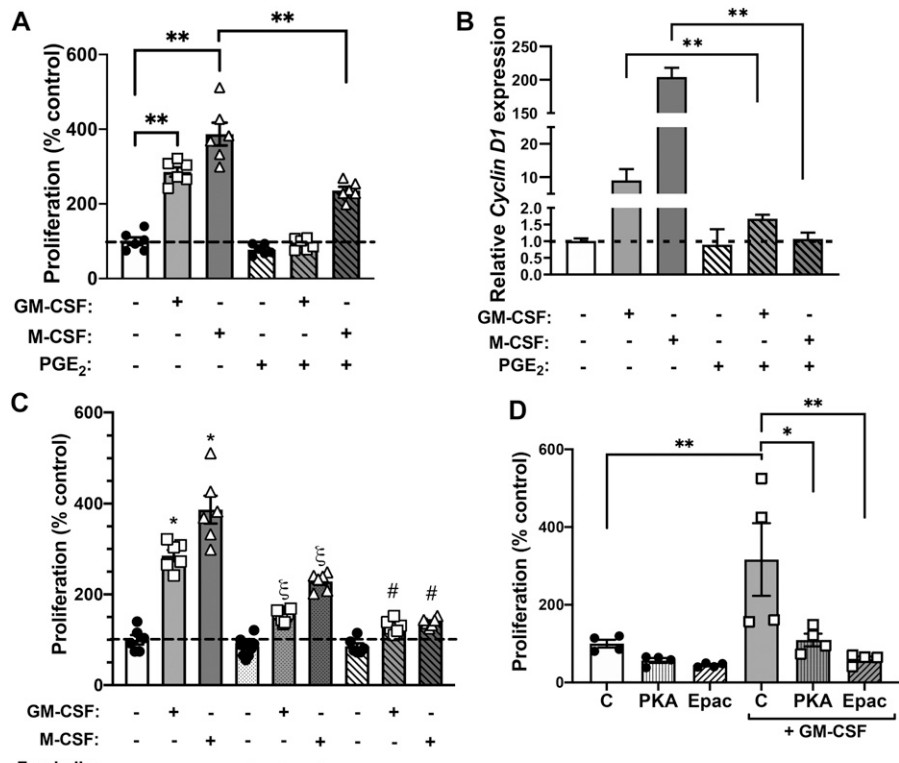

**Figure 3. PGE₂ inhibits alveolar macrophage (AM) proliferation in response to mitogens via EP2-cAMP signaling.**
**(A)** Primary mouse AMs were cultured in the presence of M-CSF or GM-CSF (10 ng/ml) with or without PGE₂ (1 μM), and their proliferation was determined by CyQuant assay measuring total cellular DNA after 5 d (n = 6 mice). **(B)** Expression of *Cyclin D1* mRNA measured by RT-PCR in primary AMs treated with M-CSF or GM-CSF with or without PGE₂ for 48 h (n = 3 separate experiments). **(C)** Proliferation of primary AMs cultured for 5 d in the presence of either M-CSF or GM-CSF in the presence of forskolin (100 μM) or the EP2 agonist butaprost (1 μM) for over 5 d. * = *P* < 0.05 compared with untreated control; § = *P* < 0.05 compared with forskolin alone; # = *P* < 0.05 compared with butaprost alone (n = 6 mice). **(D)** Proliferation of primary AMs cultured for 5 d in the presence of either M-CSF or GM-CSF in the presence of agonists for PKA (6-bnz-cAMP; 100 μM) and Epac-1 (8-pCPT-2′-O-Me-cAMP; 100 μM) (n = 4 mice). **(A, B, C, D)** Results are expressed relative to values of untreated control cells, and are shown as mean ± SEM; *P < 0.05, **P < 0.01; one-way ANOVA followed by Sidak's test for multiple comparisons (A, B, C, D).

mouse models of both aging and CS. We compared the effects of BALF from aged and young wild-type (WT) mice on proliferation of young AMs from both WT and EP2 KO mice. The ability of BALF from aged mice to suppress proliferation of AMs from WT mice was lost when the responder AMs were from EP2 KO mice (Fig 4D). This finding indicates that the effect of the suppressive factor within aged BALF—shown to contain increased levels of PGE₂ (Fig 1E and F)—depends on EP2 signaling within AMs and thus is unequivocally identified as PGE₂. Furthermore, the increase in AMs within lung digests in response to CS exposure was further exaggerated in EP2 KO mice relative to WT mice (Fig 4E), in the absence of any significant increase in neutrophils, recruited monocytes, or dendritic cells (Fig 4F). The increase in AMs in mice exposed to CS has been linked with increased GM-CSF expression in the lung (Suzuki et al, 2020). By reducing lung PGE₂ concentration (Fig 2D), smoke exposure disrupts the physiologic brake on AM proliferative self-renewal, thereby mimicking the ability of EP2 deletion (Fig 4E) to yield exaggerated AM proliferation in response to mitogenic stimuli.

## Discussion

Studies investigating the mechanisms regulating AM proliferation are limited, and potential brakes on AM proliferation are poorly understood. One recent study implicated a suppressive role for vitamin D₃ in GM-CSF–induced AM proliferation (Hu et al, 2019) that appears to play a role in the development of emphysema. Our studies revealed PGE₂ to be a novel endogenous brake on AM proliferation. Bidirectional perturbations in the lung levels of this lipid mediator contributed to alterations in AM-proliferative self-renewal in two clinically relevant models of lung immune dysfunction—aging and CS exposure.

In the instance of aging, PGE₂ levels were increased and correlated with decreased numbers of AMs within the lungs of mice of advanced age. This ability of PGE₂ to restrain AM self-renewal is consistent with its ability to inhibit proliferation of a broad variety of other cell types, including both immune (Gualde & Goodwin, 1982; Maric et al, 2018) and nonimmune (Moore et al, 2003; Huang et al, 2007; Michael et al, 2019) cells. Although PGE₂ has not, to our knowledge, been previously reported to inhibit proliferation of AMs or of any tissue macrophage population, such an effect is consistent with its ability to inhibit virtually all activation parameters of AMs, including phagocytosis, microbial killing, and inflammatory cytokine production (Aronoff et al, 2004, 2012; Serezani et al, 2007; Bourdonnay et al, 2012). Excess PGE₂ production by macrophages has been implicated in immune senescence (Wu & Meydani, 2014), although not specifically in the lung.

In contrast to the findings in aged mice, CS exposure resulted in significantly greater AM numbers in association with decreased levels of PGE₂ within the lung. The decline in PGE₂ levels was manifested within several days, reached their maximum by 7 d, and

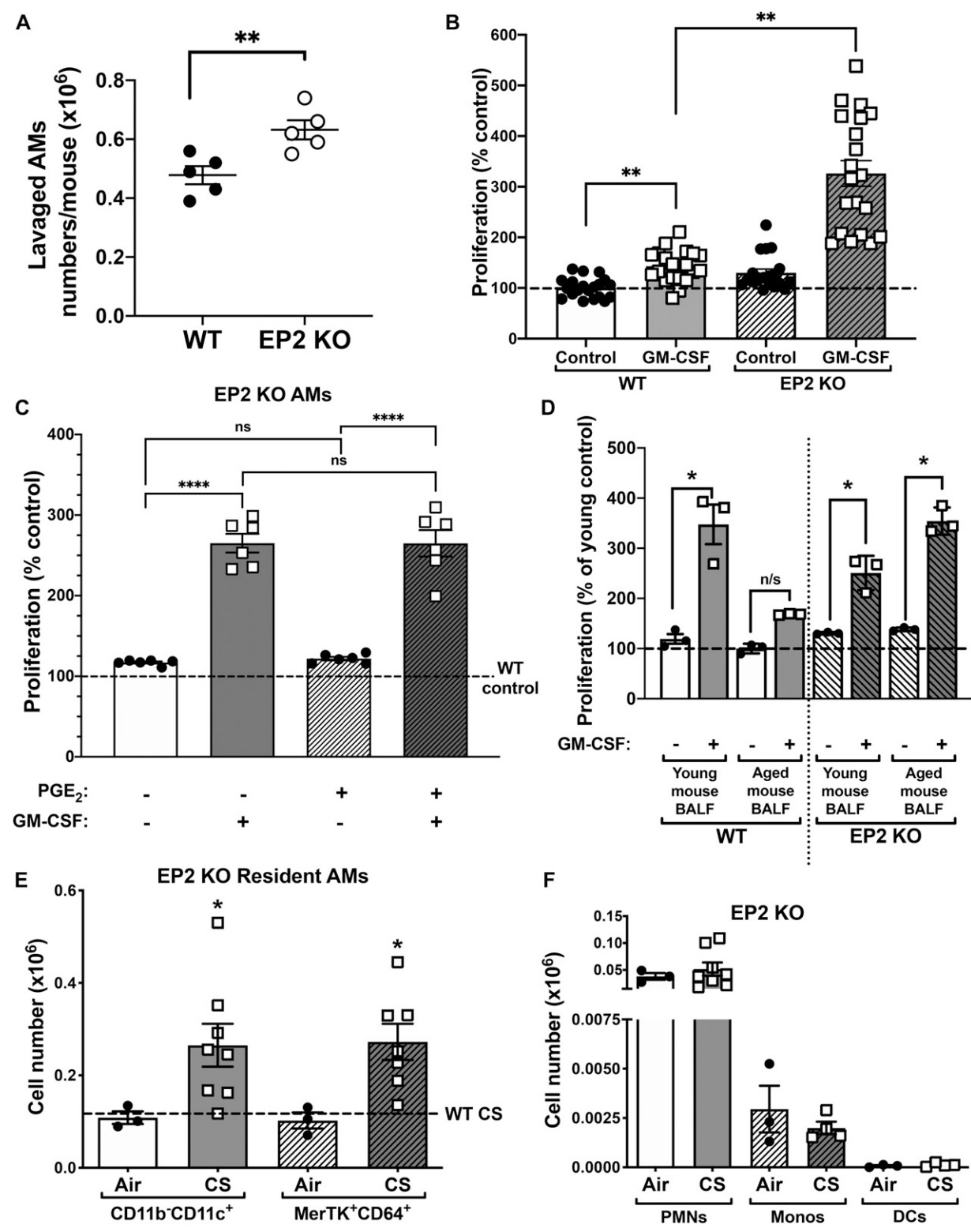

**Figure 4. Loss of EP2 signaling enhances alveolar macrophage (AM) proliferation.**
**(A)** Absolute numbers of AMs isolated from the BALF of naïve WT or EP2 KO mice (n = 5 mice). **(B)** Primary AMs were isolated from the BALF of WT and EP2 KO mice and cultured with GM-CSF (10 ng/ml) or medium alone for 5 d and then analyzed by CyQuant proliferation assay measuring total cellular DNA, using untreated WT as a control (n = 20 mice). **(C)** Proliferation of primary EP2 KO AMs cultured for 5 d with $PGE_2$, GM-CSF, or both. Values are expressed relative to untreated KO control cells. Dashed line represents WT baseline proliferation. (n = 6 mice). **(D)** Proliferation of primary young WT or EP2 KO AMs cultured for 5 d with WT young or aged BALF in the presence or absence of GM-CSF (n = 3 separate experiments; AMs from two individual mice per experiment). Values are expressed relative to untreated control for young or aged BALF,

persisted through 8 wk. The mechanism(s) by which CS exposure reduced BALF levels of $PGE_2$ was not addressed in our study. Whether CS targets AMs, alveolar epithelial cells or both cell types is uncertain. Moreover, $PGE_2$ levels reflect the balance of its synthesis, mediated by enzymes including cyclooxygenases and PGE synthases, and its degradation, mediated by 15-hydroxyprostaglandin dehydrogenase, and both classes of enzymes are subject to modulation (Tai et al, 2002; Stolberg et al, 2014). Finally, CS contains more than 5,000 discrete chemical compounds (Talhout et al, 2011), any of which, alone or in combination, could be responsible. Future research will be needed to explore the responsible compounds as well as target enzymes and cell types by which CS diminishes alveolar levels of $PGE_2$. Likewise, the mechanism(s) responsible for the elevated $PGE_2$ levels in the alveolar space of aged mice remains to be determined. Notably, the consequences of excessive (aging) or impaired (smoking) $PGE_2$ production in the lung would be expected to extend beyond regulation of AM numbers and could contribute to global impairment and activation of AM functions in aging and smoking, respectively.

Both gain of function studies with the EP2-selective agonist butaprost and the direct adenylyl cyclase activator forskolin and loss-of-function studies employing cells from EP2 KO mice pointed to a critical role for EP2 signaling via cAMP in mediating the suppressive actions of $PGE_2$ on AM proliferation in vitro and in vivo. The similar magnitude of inhibition exerted by butaprost and forskolin suggests that EP2 agonism was sufficient to capture the full potential of cAMP-mediated inhibition of AM proliferation; this in turn argues against a significant role in this inhibition for the alternative stimulatory G protein–coupled $PGE_2$ receptor, EP4. Mechanistically, studies with PKA- and Epac-selective agonists revealed that both of these cAMP effectors were sufficient to account for inhibition of proliferation. In previous work, we have reported that roles for PKA and Epac-1 in regulating other AM functions can be either distinct or redundant (Aronoff et al, 2005), a phenomenon also demonstrated in other cell types (Cheng et al, 2008).

Our in vivo findings with EP2 KO mice suggest that endogenous levels of $PGE_2$ on the respiratory surface—which could derive from both AMs and alveolar epithelial cells—are sufficient to act via EP2 on AMs to restrain their numbers. Alternatively, our in vitro finding that EP2 KO AMs demonstrate enhanced ex vivo proliferation in response to GM-CSF can only be explained if AMs themselves are the source of $PGE_2$, which in turn acts via EP2 as an autocrine brake on mitogen-stimulated proliferation. This, in turn, also suggests the possibility that the cell-intrinsic defect depicted in Fig 1B could also reflect excess autocrine production of $PGE_2$ by cultured AMs from aged mice. We speculate that a defect such as this that persists in culture may be mediated by epigenetic mechanisms; testing this possibility will require additional studies.

Although there are undoubtedly a number of determinants of AM self-renewal, our studies highlight the significance of $PGE_2$ as an endogenous suppressor of AM proliferation. Although we have

localized the $PGE_2$ effect to signaling via the EP2 receptor and downstream cAMP effectors, the molecular pathways involved in AM proliferation that are targeted for inhibition by $PGE_2$-cAMP remain to be determined. However, it is interesting that $PGE_2$ is capable of inhibiting mTORC1—a critical driver of macrophage proliferation (Okunishi et al, 2014). Further studies will be necessary to determine the distinct mechanism(s) by which $PGE_2$ exerts its effects and whether alterations in its production or in EP2 expression influence AM self-renewal in other models of lung disease.

# Materials and Methods

## Animals

For isolation of naïve AMs, 6–8-wk-old WT C57Bl/6 female mice (Jackson Laboratory) were used. For CS experiments, 6–8-wk-old mice harboring a targeted deletion of both alleles of the *Ptger2* gene encoding the EP2 receptor (EP2 KO) were used. These mice were originally provided by Dr. Richard Breyer (Vanderbilt University) (Kennedy et al, 1999) and were maintained/bred by the University of Michigan Unit of Laboratory Medicine. For aging experiments, 8-wk-old (young) and 18–22-mo-old (aged) female C57Bl/6 mice were procured from Charles River or the National Institute of Aging Animal Facility, respectively. All mice were maintained at the University of Michigan Unit for Laboratory Animal Medicine. Animals were treated according to National Institutes of Health guidelines for the use of experimental animals with the approval of the University of Michigan Committee for the Use and Care of Animals.

## AM isolation and culture

Primary AMs were lavaged from mouse lungs as previously described (Bourdonnay et al, 2012). AMs were adhered for at least 1 h in RPMI with penicillin/streptomycin (pen/strep) and 2% FBS before treatment with appropriate compounds for indicated times.

## Quantitative real-time PCR

Total RNA was extracted from $0.5 × 10^6$ AMs using QIAGEN columns per manufacturer's instructions and converted to cDNA via reverse transcription. Relative gene expression of mouse cyclin B1 and cyclin D1 was determined by the $\Delta Ct$ method, using SYBR Green dye (Applied Biosystems). Mouse *β-actin* was used as a reference gene. Primer sequences for mouse *Cyclin B1*, *Cyclin D1*, and *β-actin* were 5′-GAACCAGAGGTGGAACTTGC-3′ (f), 5′-AGATGTTTCCATCGGGCTTG-3′ (r); 5′-AGTGCGTGCAGAAGGAGATT-3′ (f), 5′-AGGAAGCGGTCCAGGTAGTT-3′ (r); and 5′-GACGGCCAGGTCATCACTAT-3′ (f), 5′-GCACTGTGTTGGCATA-GAGG-3′ (r), respectively.

---

in both WT and KO cells. **(E)** Absolute numbers of $CD11b^-CD11c^+$ and $MerTK^+CD64^+$ resident AMs isolated from lung digests of EP2 KO mice exposed to air or cigarette smoking (n = 3–8 mice) measured by flow cytometry. Dashed line represents the average absolute number of all WT resident AMs for comparison. *$P$ < 0.05 compared with air control. **(F)** Absolute numbers of $CD11b^+$ $Ly6G^+$ neutrophils, $CD11b^+Ly6C^+$ monocytes, and $CD11c^+CD103^+$ dendritic cells isolated from lung digests of EP2 KO mice exposed to air or cigarette smoking (n = 3–8 mice) measured by flow cytometry. **(A, B, C, D, E)** Results are shown as mean ± SEM; ns, nonsignificant, *$P$ < 0.05, **$P$ < 0.01; $t$ test (A) one-way ANOVA followed by Sidak's test for multiple comparisons (B, C, D, E).

## Flow cytometric analysis of resident AMs

Lungs from mice subjected to CS or air were digested with collagenase as described (Stolberg et al, 2014) and single cell suspensions were labeled with antibodies for CD11c-V450 (clone N418; eBioscience), CD11b-AlexaFluor 700 (clone M1/70; eBioscience), MerTK-FITC (clone 2B10C42; BioLegend), CD64-APC (clone X54-5/7.1; BioLegend), Ly6C-PE/Cy7 (clone HK1.4; BioLegend), Ly6G-PE (clone 1A8; BioLegend), and CD103-PerCP/Cy5.5 (clone 2E7; BioLegend) at 4°C for 1 h. The population frequencies of resident AMs (CD11b$^-$CD11c$^+$ or MerTK$^+$CD64$^+$), neutrophils (CD11b$^+$ Ly6G$^+$), recruited monocytes (CD11c$^-$CD11b$^+$Ly6C$^+$), and dendritic cells (CD11c$^+$CD11b$^-$CD103$^+$) were analyzed on a BD Fortessa flow cytometer using BD FACSDiva software.

## Proliferation assay

Primary AMs were collected from mouse lung lavage fluid as described above. Cells were adhered in RPMI with 2% FBS and pen/strep in a 96-well plate at a concentration of 5,000 cells/well. Cells were treated with mitogen (mouse rM-CSF [10 ng/ml] or mouse rGM-CSF [10 ng/ml]; Peprotech) and incubated for 5 d. In some experiments, cells were pretreated with 1 $\mu$M PGE$_2$ (Cayman Chemical), 100 $\mu$M forskolin (Calbiochem), 1 $\mu$M EP2-selective agonist (butaprost; Cayman Chemical), 100 $\mu$M PKA agonist (6-bnz-cAMP; Axxora), or 100 $\mu$M Epac-1 agonist (8-pCPT-2′-O-Me-cAMP; Axxora) for 1 h before mitogen stimulation. Cell proliferation was determined after 5 d of culture using the CyQuant proliferation assay (Thermo Fisher Scientific) for DNA binding as per the manufacturer's instructions.

## Lipidomic analysis

BALF samples (800 $\mu$l) from young and aged mice (3–4 mice/group) were analyzed for lipid mediator content using an eicosanoid 44-plex lipidomic panel by Cayman Chemical. Briefly, BALF was subjected to liquid-phase and solid-phase extraction before immunoaffinity capture, then run through liquid chromatography and tandem mass spectrometry (LC instrument: Sciex ExionLC Integrated System, MS instrument: Sciex Triple Quad 6500+). Calculations of the total amount of each lipid present in the samples were performed using MultiQuant software (Sciex).

## PGE$_2$ ELISA

Individual mouse lungs were flushed with 800 $\mu$l PBS. The resulting lavage fluid was centrifuged at 500$g$ for 10 min at 4°C to remove dead cells and debris, and PGE$_2$ levels were assessed by ELISA (Enzo Life Sciences).

## CS exposure

For in vivo CS exposure, mice were exposed to mainstream and side-stream CS as previously described (Phipps et al, 2010). Briefly, smoke from standardized 2R4F research cigarettes (University of Kentucky) was generated by a TE-2 CS machine (Teague Enterprises). Animals were exposed for 2 h/d and 7 d/wk in a whole-body exposure chamber. The mean concentration of particulates collected during a 2 h exposure was 19.05 ± 3.96 mg/m$^3$/d. Control animals were housed in an identical chamber but were exposed to room air with no smoke.

## EdU incorporation in CS-exposed AMs

For assessment of resident AM proliferation in response to CS exposure, mice were injected i.p. with 1 mg/kg EdU (BaseClick GmbH) immediately before CS exposure every other day for a total of four injections, starting 1 d before to the first CS exposure. Mice were euthanized within 2 h of final CS exposure and AMs were isolated from BALF as described previously (Bourdonnay et al, 2012). AMs from individual mice were labeled with fluorescently conjugated antibodies for CD11c and CD11b (sources as noted above), before fixation and permeabilization with BD CytoFix/CytoPerm reagent (BD Biosciences). EdU fluorescence was initiated using an in vivo EdU kit (BaseClick GmbH) per the manufacturer's instructions. CD11c$^+$CD11b$^-$EdU$^+$ AMs were enumerated via flow cytometry.

## Statistics

Data are presented as mean ± SEM unless otherwise specified. Differences between groups were tested using a one-way ANOVA followed by Sidak's test for multiple comparisons, or by a $t$ test, as appropriate, using Prism 8.0 (GraphPad Software). For correlation between AM numbers and PGE$_2$ levels within the lungs of young and aged mice, Pearson's correlation analysis was performed using matched samples from 8 to 10 mice per group. $P$-values below 0.05 were considered to be significant.

# Supplementary Information

# Acknowledgements

The authors would like to thank Rachel L Zemans for her assistance with the EdU incorporation assay procedure and members of the Peters-Golden laboratory for helpful input. The studies in this manuscript were supported by National Institues of Health Grants R01 HL125555 and R35 HL144979 (to M Peters-Golden) and National Institutes of Health grants R01AG028082, R01HL120669, R01HL120669, R01AI138347, and K07AG050096 (to DR Goldstein). JM Speth and C Draijer were supported by National Institutes of Health T32 training grant HL 7749-23, JM Speth was also supported by Michigan Institute for Clinical & Health Research (MICHR) grant UL1TR00043 and American Cancer Society grant PF-17-143-01-TBG3. Z Zaslona was supported by an American Lung Association senior research fellowship award. P Mancuso was supported by Flight Attendants Medical Research Institute grant CIA-103071. J Chen is supported by National Institutes of Health T32 training grant T32AI007413. JL Curtis and CM Freeman were supported by Merit Review awards I01 CX000911 and I01 CX001553, respectively, from the Clinical Science Research & Development Service, Department of Veterans Affairs.

## Author Contributions

LR Penke: conceptualization, resources, data curation, formal analysis, validation, investigation, methodology, project administration, and writing—original draft, review, and editing.

JM Speth: conceptualization, resources, data curation, formal analysis, validation, investigation, methodology, project administration, and writing—original draft, review, and editing.

C Draijer: conceptualization, data curation, and validation.

Z Zaslona: conceptualization, data curation, and validation.

J Chen: data curation, validation, investigation, methodology, and writing—original draft.

P Mancuso: data curation, validation, investigation, and methodology.

CM Freeman: conceptualization, resources, data curation, and writing—original draft.

JL Curtis: conceptualization, data curation, investigation, and writing—original draft.

DR Goldstein: conceptualization, data curation, formal analysis, supervision, funding acquisition, validation, investigation, and writing—original draft.

M Peters-Golden: conceptualization, formal analysis, supervision, funding acquisition, validation, investigation, project administration, and writing—original draft, review, and editing.

## Conflict of Interest Statement

The authors declare that they have no conflict of interest.

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
