## [Reviewer comments · Life Science Alliance]

Life Science Alliance

PGE2 accounts for bidirectional changes in alveolar macrophage self-renewal with aging and smoking

Loka Penke, Jennifer Speth, Christina Draijer, Zbigniew Zaslona, Judy Chen, Peter Mancuso, Christine Freeman, Jeffrey Curtis, Daniel Goldstein, and Marc Peters-Golden

DOI: <https://doi.org/10.26508/lsa.202000800>

Corresponding author(s): Marc Peters-Golden, University of Michigan

Review Timeline:

Submission Date:	2020-05-29
Editorial Decision:	2020-07-14
Revision Received:	2020-08-12
Editorial Decision:	2020-08-13
Revision Received:	2020-08-13
Accepted:	2020-08-14

Scientific Editor: Shachi Bhatt

Transaction Report:

July 14, 2020

Re: Life Science Alliance manuscript #LSA-2020-00800-T

Prof. Marc Peters-Golden
University of Michigan
Pulmonary and Critical Care Medicine
1150 west medical center drive
Ann Arbor, Michigan 48109

Dear Dr. Peters-Golden,

Thank you for submitting your manuscript entitled "PGE2 accounts for bidirectional changes in alveolar macrophage self-renewal with aging and smoking" to Life Science Alliance. The manuscript was assessed by expert reviewers, whose comments are appended to this letter.

We share the referees enthusiasm for your dataset and would warmly welcome the submission of a revised manuscript for publication in LSA.

Please address Referee #1's comments by performing the experiments suggested by him/her, principally, to determine the extent to which unopposed EP1,3 or 4 effects EP2 deficiency on macrophage numbers, to determine if there is a correlation between levels of PGE2 and the macrophage numbers in aged mice and determine if newly proliferated macrophages are producers of PGE2.

Please include Referee #2's recommendations for textural changes.

Please comment on Referee 3's query on the BAL experiment re. cultures that did not apparently include PGE2.

We will not require data on ATACseq additional human data.

In our view these revisions should typically be achievable in around 3 months. However, we are aware that many laboratories cannot function fully during the current COVID-19/SARS-CoV-2 pandemic and therefore encourage you to take the time necessary to revise the manuscript to the extent requested above. We will extend our 'scoping protection policy' to the full revision period required. If you do see another paper with related content published elsewhere, nonetheless contact me immediately so that we can discuss the best way to proceed.

Please let us know if you prefer to discuss any of the revision recommendations directly, in particular if your laboratory suffers from COVID-19 related restrictions. We would be happy to discuss the individual revision points further with you.

Please note that papers are generally considered through only one revision cycle, so strong support from the referees on the revised version is needed for acceptance.

Thank you for this interesting contribution to Life Science Alliance. We are looking forward to receiving your revised manuscript.

Sincerely,

Reilly Lorenz
Editorial Office Life Science Alliance
Meyerhofstr. 1
69117 Heidelberg, Germany
t +49 6221 8891 414
e contact@life-science-alliance.org
www.life-science-alliance.org

B. MANUSCRIPT ORGANIZATION AND FORMATTING:

Reviewer #1 (Comments to the Authors (Required)):

General comments

This is a careful study of the role of PGE2 in determining the numbers of resident alveolar macrophages in mice that are exposed to cigarette smoke and aged mice. It is concluded based on pharmacological data that EP2, a PGE2 receptor normally maintains a lower number of alveolar macrophages than is the case in the absence of the receptor and that differences in PGE2 levels in the alveolar compartment with cigarette smoking and aging determine the observed differences in macrophages. The data are convincing although the mechanisms leading to the differences in PGE2 in the two circumstances are not known or revealed in the current paper.

Major comments

To what extent does unopposed EP1,3 or 4 determine the effects of EP2 deficiency on alveolar macrophage numbers? Perhaps some discussion of this issue would be useful.

Line 158. Is there evidence that nicotine can downregulate cyclooxygenase or Ptges in these cells? The mechanism is still lacking as to why PGE2 synthesis is altered? It would be good to add something to the discussion if possible.

Figure 1, panel F shows a great variance of PGE2 levels in BALF of aged mice. Indeed half of the data fall within the normal range. Is there any correlation between the levels observed and the macrophage numbers in the aged mice?

Figure 2, panel E. Are the newly proliferated macrophages poor producers of PGE2? It does seem that the effect takes a while to be seen, suggesting a dependence on the expansion of the macrophage population.

Figure 4, panel B. It is not clear to this reviewer why the WT AMs did not proliferate with GM-CSF.

Minor comments

The summary of the paper is somewhat unclear. Please re-phrase in more explicit terms.

Line 45. Why a "tunable" brake? Is the evidence that PGE2 changes the basis for calling it a tunable brake?

Line 204. Aging is perhaps not correctly described as a model of lung dysfunction.

Reviewer #2 (Comments to the Authors (Required)):

This manuscript represents an extension to previous work published by this group on the role of prostaglandin E2 (PGE2) in altering macrophage effector mechanisms. They present two models: cigarette smoke and ageing. In the cigarette smoke model macrophages increase and in ageing they decrease. They correlate these changes with fluctuations in PGE2 that signals via the EP2 receptor.

I expect that this manuscript has been criticised in the past by other studies that promote a definitive role for other factors such as GM-CSF in macrophage renewal in the respiratory tract. The

likelihood is that many factors affect macrophage renewal versus replacement by circulating monocytes. Difficulties arise when studies imply a singular role for the mechanism under study. To some extent this manuscript does this, and it would be beneficial for the authors to tone down the dominance of their proposed pathway and acknowledge that airway macrophages renewal must be regulated by multiple pathways.

The authors should also acknowledge that, for the most part, they have only studied one time point and that aged macrophages in particular may simply be slower in responding to PGE2.

Sometimes the presentation of figures leads to misunderstandings as the legend is not clear what they are comparing their results with. In figure 1c for example, how was the "relative" data reached. Sometimes they examine fold change of old macrophages in response to stimuli by looking at fold changes above young macrophages. It would make more sense to compare with unstimulated aged macrophages. Either way, the legends need to provide clarity.

In cigarette smoke the monocytes do increase and so the authors should acknowledge that these might help increase the macrophage pool.

The authors should further acknowledge that the fact that EP2 blockers, replicate their results, this does not necessarily mean it is the only pathway and that blocking others, including EP4 may do the same.

Figure 1 could be omitted as it largely repeats their published data

Reviewer #3 (Comments to the Authors (Required)):

In this manuscript entitled "PGE2 accounts for bidirectional changes in alveolar macrophage self-renewal with aging and smoking" Penke and colleagues have investigated the potential contribution of PGE2 in AM proliferation during physiological (aging) and pathological (smoking) in a mouse model. The manuscript has a clear hypothesis with well-executed experiments to test it. By using loss-and-gain of function approaches, the authors concluded that PGE2 is negatively regulate AM proliferation that may explain the susceptibility of aged people to pulmonary infections or the pathological link between excessive AMs and COPD. This is a very important observation and due to availability of multiple pharmacological inhibitors targeting PGE2 pathways, the study has tremendous clinical implication. I have only two minor comments:

1) As the ex vivo study was performed in culture that requires BAL and isolation of cells from the lung microenvironment containing PGE2, how these cells in the culture system that has no PGE2 maintain their phenotype? If the authors envision that there should be a PGE2-mediated epigenetic imprinting that intrinsically maintain their phenotype, how long this imprinting will last independent of PGE2? I think, if the authors perform ATACseq experiment prior to culture and 5 days after culture, this will significantly improve the conceptual and the potential mechanisms involved in this observation. Does aging or smoking has any impact on BMDM function under any of these conditions?

2) The manuscript will tremendously benefit from human data demonstrating the levels of PGE2 in BAL of aged/smoker/COPD. While the authors provide Refs of potential pathological link between PGE2 and the number of AMs in smokers/COPD, is there any possibility to consider that this

increased number of AM is linked to enhance host defense against pulmonary infections? While a previous study by this group has shown that aging impairs AM phagocytosis and increased susceptibility to IAV infection due to decreased disease tolerance (Wong et al. JI, 2017), another study demonstrated that PGE2 regulates host resistance via AM/type I IFN axis (Coulombe et al. Immunity, 2014). For insatnce, does aging or smoking has any impact on type I IFN signaling in AM?

We thank the reviewers for their overall positive assessment of our study as well as their thoughtful and constructive comments. We will address the reviewers' comments on a point-by-point basis. Changes made to the revised manuscript are indicated therein by the highlighted text. We believe that these modifications have improved the manuscript and hope that it is now appropriate for publication.

Reviewer #1:

To what extent does unopposed EP1,3 or 4 determine the effects of EP2 deficiency on alveolar macrophage numbers? Perhaps some discussion of this issue would be useful.

Our previous work suggests that the EP2 receptor is the predominant EP receptor expressed by AMs and mediates most of PGE₂'s suppressive actions in this cell type (Aronoff et al., 2004; Medeiros et al., 2009). However, we agree with the reviewer's comment about the possibility of PGE₂ utilizing other EP receptors to drive AM proliferation; in such a scenario, EP1 or 3 would be most likely as these signal via stimulatory pathways whereas EP4, like EP2, signals via cAMP. To investigate this, we conducted new experiments in which we harvested EP2 KO AMs and treated them with PGE₂ to determine its ability to influence AM proliferation in the absence of suppressive EP2. As shown in our revised Figure 4 (panel 4C), these new data indicate that PGE₂ alone had no appreciable effect on AM proliferation in EP2 KO AMs either in the presence or absence of concomitant GM-CSF. This excludes the possibility that PGE₂ itself via EP receptors 1,3 or 4 is involved in the enhanced AM proliferation observed in EP2 KO cells. Rather, this increased proliferation is the consequence of removing the EP2 brake. **We have expanded our discussion of this point in the text (Lines 191-200; 257-263).**

Is there evidence that nicotine can downregulate cyclooxygenase or Ptges in these cells? The mechanism is still lacking as to why PGE2 synthesis is altered? It would be good to add something to the discussion if possible.

We acknowledge that the current manuscript is lacking in mechanistic interrogation regarding the reduction in PGE₂ levels in mice exposed to cigarette smoke, as it is for the increase in PGE₂ observed in aged mice. Cigarette smoke contains not only nicotine but thousands of other compounds. In an effort to probe possible mechanisms, we performed in vitro experiments using cigarette smoke extract (CSE). However, we found significant toxicity and cell death in AMs treated with CSE, even at very low concentrations. Therefore, we abandoned this effort and limited our studies to in vivo cigarette smoke exposure. It should also be recognized that the AM is not the only cell on the alveolar surface capable of contributing PGE₂ to levels in the BALF, and there is in fact evidence to suggest that the epithelium is likely its predominant source (Chauncey et al, 1988; Lipchik et al, 1990). Finally, the degradation of PGE₂ by the enzyme 15-PGDH is also subject to modulation, and an increase in its activity – with or without concomitantly reduced cyclooxygenases or PGE synthases – could contribute to the reduction with smoking. Thus, the reduction in BALF levels of PGE₂ may reflect complex interactions between numerous components of CSE and various enzymes in a variety of lung cell types. **As suggested by the reviewer, we have acknowledged this open question and elaborated upon this issue in our revised manuscript (lines 242-250).**

Figure 1, panel F shows a great variance of PGE2 levels in BALF of aged mice. Indeed half of

the data fall within the normal range. Is there any correlation between the levels observed and the macrophage numbers in the aged mice?

We acknowledge the variability of BALF PGE₂ levels evident in our previously presented data with aged mice and the overlap in PGE₂ levels with those in young mice. Unfortunately, in these previous data we failed to record the AM numbers to match the individual PGE₂ values from a given mouse. We therefore performed such a correlation analysis between PGE₂ levels and AM numbers in a new cohort of both young and aged mice. We incorporated the additional PGE₂ and AM number values into the data depicted in Figure 1A and F, and new Figure 1G presents the correlation analysis from this new experiment (Lines 117-121). These data demonstrate a moderate but significant negative correlation between AM number and PGE₂ levels.

Figure 2, panel E. Are the newly proliferated macrophages poor producers of PGE2? It does seem that the effect takes a while to be seen, suggesting a dependence on the expansion of the macrophage population.

While it is possible that newly proliferated AMs are inferior producers of PGE₂, we have not explored this. As noted above, it is also possible (and we suspect more likely) that this reflects impaired PGE₂ synthesis by epithelial cells. We are currently investigating the effects of cigarette smoke exposure on alveolar epithelial cells, which we believe to be a separate story from the current manuscript. Again, we have acknowledged in the revised Discussion that the mechanism for this decline remains to be determined.

Figure 4, panel B. It is not clear to this reviewer why the WT AMs did not proliferate with GM-CSF.

We thank the reviewer for pointing out this obvious discrepancy. Upon further investigation, we realized that the control data were improperly analyzed. We have performed a new analysis with statistics and have updated the Figure 4B to reflect the correct data values.

The summary of the paper is somewhat unclear. Please re-phrase in more explicit terms.

We regret that the reviewer was dissatisfied with our summary, but without more guidance or detail, we are unsure as to what the reviewer means by “please re-phrase in more explicit terms”.

Line 45. Why a "tunable" brake? Is the evidence that PGE2 changes the basis for calling it a tunable brake?

Yes, we meant that PGE₂ is a brake whose “force” (i.e., concentration level) can be tuned up or down by virtue of modulation of the enzymes responsible for its biosynthesis and/or degradation. In aging, PGE₂ levels are enhanced (the brake is increased), resulting in greater inhibition of AM proliferation. In cigarette smoke exposure its levels are reduced (the brake is removed), resulting in more AM proliferation. We believe the term “tuneable” is suitable to describe the capacity to modulate its levels depending on the environment.

Line 204. Aging is perhaps not correctly described as a model of lung dysfunction.

Of course aging is a normal phenomenon! We have reworded this to clarify that aging is a model in which there is dysfunction of the immune system of the lung (Lines 160-161; 224-225).

Reviewer #2:

.... Difficulties arise when studies imply a singular role for the mechanism under study. To some extent this manuscript does this, and it would be beneficial for the authors to tone down the dominance of their proposed pathway and acknowledge that airway macrophages renewal must be regulated by multiple pathways.

We agree with the reviewer that PGE₂ is unlikely to be the only mediator involved in the suppression of AM proliferation. We have made an effort to change the wording within the revised manuscript to be less singular and have discussed other potential mechanisms within the discussion (Line 280; 220-222).

The authors should also acknowledge that, for the most part, they have only studied one time point and that aged macrophages in particular may simply be slower in responding to PGE₂.

We are unsure what the reviewer means by “one timepoint” in regards to our aged mouse model. We assume they are referring not to the age of the “aged” mice, but rather to the fact that we measured proliferation exclusively during a 5-day culture time with mitogens. In our previous studies (Draijer et al, 2019), we found that 5 days is the optimal time in which we see AM proliferation in response to mitogens such as GM-CSF. If we were studying a rapid signaling response (such as a change in calcium or kinase activation) (e.g., over 10-60 minutes), one could certainly imagine that cells from aged mice were simply more sluggish and a longer time point (e.g., 30-120 minutes) might be required to demonstrate this. However, we feel that 5 days is a sufficiently long proliferation interval to capture a slower response even if it was manifest by cells from aged mice.

Sometimes the presentation of figures leads to misunderstandings as the legend is not clear what they are comparing their results with. In figure 1c for example, how was the "relative" data reached. Sometimes they examine fold change of old macrophages in response to stimuli by looking at fold changes above young macrophages. It would make more sense to compare with unstimulated aged macrophages. Either way, the legends need to provide clarity.

We thank the reviewer for their input. We have revised the legends and attempted to make them more clear and concise.

In cigarette smoke the monocytes do increase and so the authors should acknowledge that these might help increase the macrophage pool.

While it appears that monocyte numbers increase after cigarette smoke exposure, this increase is not statistically significant. Moreover, it is not likely that the increase in absolute numbers of monocytes (~0.0005x10⁶ cells) would meaningfully contribute to the much greater increase in resident AM numbers (~0.1x10⁶ cells). We have addressed this in Lines 138-140 in the text.

The authors should further acknowledge that the fact that EP2 blockers, replicate their results, this does not necessarily mean it is the only pathway and that blocking others, including EP4 may do the same.

In this study, we used agonists for cAMP (forskolin) and EP2 (butaprost) as well as EP2 receptor knockout mice to identify and confirm the PGE₂ signaling pathway responsible for inhibition of AM proliferation. We did not perform studies using antagonists to block EP2, and we assume that reviewer incorrectly used the term “blocker” instead of agonist. As suggested by the reviewer, EP4 is also known to signal via cAMP, and could theoretically mediate some of the suppressive actions of PGE₂ on AM proliferation. However, the data in Figure 3C show that the selective EP2 agonist butaprost is at least as suppressive as the adenylyl cyclase activator forskolin, and this suggests that EP2 agonism is sufficient to capture the full potential of cAMP-mediated inhibition. **This issue has been discussed in the revised manuscript (Lines 191-200; 257-263).**

Figure 1 could be omitted as it largely repeats their published data.

We respectfully disagree that Figure 1 deserves to be omitted. A previous publication by our group reported a decrease in AM numbers and proliferation in aged mice by analyzing lung digests and Ki67 staining of isolated AMs. Our new data extend these initial findings by measuring AM frequency in lung lavage rather than digests (1A) and directly analyzing their proliferative capacity (1B) and cell cycle gene expression(1C) in response to mitogenic stimuli; we believe that validation of this reduction in AM numbers in aged mice and further exploration of mechanisms are important for the field. Moreover, the BALF swap experiment (1D), lipid analysis (1E), and PGE₂ ELISA data (1F) are completely novel observations. We believe strongly that omitting this figure would exclude important new information.

Reviewer #3:

As the ex vivo study was performed in culture that requires BAL and isolation of cells from the lung microenvironment containing PGE2, how these cells in the culture system that has no PGE2 maintain their phenotype?

Our proliferation assays involve culture of isolated AMs for 5 days. During this time, AMs are capable of producing their own PGE₂ which could contribute to the maintenance of their phenotype. It is also possible that these AMs manifest epigenetic changes acquired in vivo that contribute to enhanced PGE₂ production that is maintained throughout the culture period. **We discuss these possibilities in lines (270-278) of the manuscript.**

If the authors envision that there should be a PGE2-mediated epigenetic imprinting that intrinsically maintain their phenotype, how long this imprinting will last independent of PGE2? I think, if the authors perform ATACseq experiment prior to culture and 5 days after culture, this will significantly improve the conceptual and the potential mechanisms involved in this observation.

As suggested by the reviewer, ATACseq may strengthen our findings. However, we consider such an exploration to be beyond the scope of the present manuscript.

Does aging or smoking has any impact on BMDM function under any of these conditions?

We have not addressed herein the relevant question of whether aging or smoking alter the function of monocytes originating in the bone marrow which are recruited to the lung. Once monocytes are recruited to the lung, they are gradually conditioned by the components of the alveolar milieu to acquire characteristics typical of resident AMs; thus, we would speculate that alterations in the levels of PGE₂ in the alveolar milieu would influence recruited cells as it does resident AMs. Since recruited cells do not seem to significantly contribute to the expanded AM numbers in CS, we have chosen not to include this speculation in the current manuscript.

The manuscript will tremendously benefit from human data demonstrating the levels of PGE2 in BAL of aged/smoker/COPD.

We agree that human BAL data would increase the impact of our manuscript, however due to recent limitations on human subject research, we are currently unable to obtain protocols to acquire these samples. Therefore, we believe this to be beyond the scope of the current study.

While the authors provide Refs of potential pathological link between PGE2 and the number of AMs in smokers/COPD, is there any possibility to consider that this increased number of AM is linked to enhance host defense against pulmonary infections?

Increases in AM numbers with smoking have been linked to emphysema (Suzuki, et al 2020), and depletion of AMs with intrapulmonary administration of clodronate have been shown to impair host defense against various pathogens (Reed et al, 2008; Traeger et al, 2009; Murphy et al, 2011). However, we are unaware of data demonstrating a quantitative relationship between AM numbers and host immune responses. Decreases in PGE₂ would be expected to not only increase AM numbers but also bacterial phagocytosis and killing. Yet, smokers' AMs are considered to exhibit impaired antibacterial function. Clearly, there are multiple determinants of host defense beyond simply AM numbers.

While a previous study by this group has shown that aging impairs AM phagocytosis and increased susceptibility to IAV infection due to decreased disease tolerance (Wong et al. JI, 2017), another study demonstrated that PGE2 regulates host resistance via AM/type I IFN axis (Coulombe et al. Immunity, 2014). For insatnce, does aging or smoking has any impact on type I IFN signaling in AM?

PGE₂ is a tremendously pleiotropic mediator which modulates numerous components of the immune system. Inhibition of the type I IFN axis is one such effect. We have not examined the impact of either aging or smoking on this axis in AMs, as it was not our purpose to explore antiviral or other host responses. Certainly, an adequate exploration of this question would require dedicated studies which are beyond the scope of this manuscript.

August 13, 2020

RE: Life Science Alliance Manuscript #LSA-2020-00800-TR

Prof. Marc Peters-Golden
University of Michigan
Pulmonary and Critical Care Medicine
1150 west medical center drive
Ann Arbor, Michigan 48109

Dear Dr. Peters-Golden,

Thank you for submitting your revised manuscript entitled "PGE2 accounts for bidirectional changes in alveolar macrophage self-renewal with aging and smoking". We would be happy to publish your paper in Life Science Alliance pending final revisions necessary to meet our formatting guidelines.

Please make the following changes in the revised manuscript for publication,

-please add a conflict of interest statement as a separate section in your manuscript text

-please use the [10 author names, et al.] format in your references (i.e. limit the author names to the first 10)

-please consider the following edited Summary statement -

"Dysregulation of proliferative self-renewal contributes to functional alterations in alveolar macrophages (AMs). In this study, Penke et al. identify PGE2 as a suppressor of AM proliferation. The divergent bioavailability of PGE2 in aged and cigarette smoke-exposed mice could explain the opposite effects seen on AM numbers."

A. FINAL FILES:

B. MANUSCRIPT ORGANIZATION AND FORMATTING:

Sincerely,

Shachi Bhatt
Scientific Editor
Life Science Alliance

August 14, 2020

RE: Life Science Alliance Manuscript #LSA-2020-00800-TRR

Prof. Marc Peters-Golden
University of Michigan
Pulmonary and Critical Care Medicine
1150 west medical center drive
Ann Arbor, Michigan 48109

Dear Dr. Peters-Golden,

Thank you for submitting your Research Article entitled "PGE2 accounts for bidirectional changes in alveolar macrophage self-renewal with aging and smoking". It is a pleasure to let you know that your manuscript is now accepted for publication in Life Science Alliance. Congratulations on this interesting work.

DISTRIBUTION OF MATERIALS:

Again, congratulations on a very nice paper. I hope you found the review process to be constructive and are pleased with how the manuscript was handled editorially. We look forward to future exciting submissions from your lab.

Sincerely,

Shachi Bhatt
Executive Editor
Life Science Alliance